# Evaluation of Endobronchial Ultrasound-Guided Transbronchial Needle Aspiration (EBUS-TBNA) Samples from Advanced Non-Small Cell Lung Cancer for Whole Genome, Whole Exome and Comprehensive Panel Sequencing

**DOI:** 10.3390/cancers16040785

**Published:** 2024-02-15

**Authors:** David Fielding, Vanessa Lakis, Andrew J. Dalley, Haarika Chittoory, Felicity Newell, Lambros T. Koufariotis, Ann-Marie Patch, Stephen Kazakoff, Farzad Bashirzadeh, Jung Hwa Son, Kimberley Ryan, Daniel Steinfort, Jonathan P. Williamson, Michael Bint, Carl Pahoff, Phan Tien Nguyen, Scott Twaddell, David Arnold, Christopher Grainge, Andrew Pattison, David Fairbairn, Shailendra Gune, Jemma Christie, Oliver Holmes, Conrad Leonard, Scott Wood, John V. Pearson, Sunil R. Lakhani, Nicola Waddell, Peter T. Simpson, Katia Nones

**Affiliations:** 1UQ Centre for Clinical Research, Faculty of Medicine, The University of Queensland, Brisbane, QLD 4029, Australia; a.dalley@uq.edu.au (A.J.D.); h.chittoory@uq.edu.au (H.C.); s.lakhani@uq.edu.au (S.R.L.); p.simpson@uq.edu.au (P.T.S.); 2Department of Thoracic Medicine, The Royal Brisbane & Women’s Hospital, Brisbane, QLD 4006, Australia; farzad.bashirzadeh@health.qld.gov.au (F.B.); junghwa.son@health.qld.gov.au (J.H.S.); kimberley.ryan@health.qld.gov.au (K.R.); 3QIMR Berghofer Medical Research Institute, Brisbane, QLD 4006, Australia; vanessa.lakis@qimrberghofer.edu.au (V.L.); felicity.newell@qimrberghofer.edu.au (F.N.); ross.koufariotis@qimrberghofer.edu.au (L.T.K.); devonpatches@gmail.com (A.-M.P.); stephen.kazakoff@qimrberghofer.edu.au (S.K.); oliver.holmes@qimrberghofer.edu.au (O.H.); conrad.leonard@qimrberghofer.edu.au (C.L.); scott.wood@qimrberghofer.edu.au (S.W.); john.pearson@qimrberghofer.edu.au (J.V.P.); nic.waddell@qimrberghofer.edu.au (N.W.); katia.nones@qimrberghofer.edu.au (K.N.); 4Department of Respiratory and Sleep Medicine, Royal Melbourne Hospital, Melbourne, VIC 3050, Australia; daniel.steinfort@mh.org.au (D.S.); jemma.christie@mh.org.au (J.C.); 5Department of Thoracic Medicine, Liverpool Hospital Sydney, Sydney, NSW 2170, Australia; jpw@jwilliamson.com.au; 6Department of Respiratory and Sleep Medicine, Sunshine Coast University Hospital, Birtinya, QLD 4575, Australia; michael.bint@health.qld.gov.au (M.B.); andrew.pattison@health.qld.gov.au (A.P.); 7Department of Thoracic Medicine, Gold Coast University Hospital, Southport, QLD 4215, Australia; carl.pahoff@health.qld.gov.au; 8Department of Thoracic Medicine, Royal Adelaide Hospital, Adelaide, SA 5000, Australia; phantien.nguyen@sa.gov.au; 9Department of Respiratory and Sleep Medicine, John Hunter Hospital, Newcastle, NSW 2305, Australia; scott.twaddell@health.nsw.gov.au (S.T.); david.arnold@health.nsw.gov.au (D.A.); christopher.grainge@health.nsw.gov.au (C.G.); 10Pathology Queensland, The Royal Brisbane & Women’s Hospital, Brisbane, QLD 4006, Australia; david.fairbairn@health.qld.gov.au; 11NSW Health Pathology South, Liverpool Hospital, Sydney, NSW 2170, Australia; shailendra.gune@health.nsw.gov.au; 12School of Biomedical Sciences, The University of Queensland, Brisbane, QLD 4067, Australia

**Keywords:** fresh aspirates, whole genome sequencing, whole exome sequencing, TSO500, advanced lung cancer, EBUS-TBNA

## Abstract

**Simple Summary:**

EBUS-TBNA specimens are the most common source of diagnostic tissue from patients with advanced inoperable lung cancer. Genomic testing is critical to inform treatment options, and a wider adoption of comprehensive sequencing would improve the detection of actionable mutations and outcomes for patients. However, patients can miss out on genomic testing when there is insufficient sample in the EBUS-TBNA specimen. Here we evaluated the largest cohort of freshly collected EBUS-TBNA specimens and compared across three comprehensive sequencing platforms for the detection of actionable mutations and potential biomarkers of treatment responses. This study demonstrates the enormous potential of fresh EBUS-TBNA samples as important biospecimens for clinical testing and for treatment response biomarker discovery.

**Abstract:**

Endobronchial ultrasound-guided transbronchial needle aspiration (EBUS-TBNA) is often the only source of tumor tissue from patients with advanced, inoperable lung cancer. EBUS-TBNA aspirates are used for the diagnosis, staging, and genomic testing to inform therapy options. Here we extracted DNA and RNA from 220 EBUS-TBNA aspirates to evaluate their suitability for whole genome (WGS), whole exome (WES), and comprehensive panel sequencing. For a subset of 40 cases, the same nucleic acid extraction was sequenced using WGS, WES, and the TruSight Oncology 500 assay. Genomic features were compared between sequencing platforms and compared with those reported by clinical testing. A total of 204 aspirates (92.7%) had sufficient DNA (100 ng) for comprehensive panel sequencing, and 109 aspirates (49.5%) had sufficient material for WGS. Comprehensive sequencing platforms detected all seven clinically reported tier 1 actionable mutations, an additional three (7%) tier 1 mutations, six (15%) tier 2–3 mutations, and biomarkers of potential immunotherapy benefit (tumor mutation burden and microsatellite instability). As expected, WGS was more suited for the detection and discovery of emerging novel biomarkers of treatment response. WGS could be performed in half of all EBUS-TBNA aspirates, which points to the enormous potential of EBUS-TBNA as source material for large, well-curated discovery-based studies for novel and more effective predictors of treatment response. Comprehensive panel sequencing is possible in the vast majority of fresh EBUS-TBNA aspirates and enhances the detection of actionable mutations over current clinical testing.

## 1. Introduction

Targeted treatments and immunotherapy have transformed lung cancer clinical care [1]. The detection of biomarkers for treatment response plays an important role in the treatment selection, quality of life, and clinical outcome for lung cancer patients [2]. Acquiring tumor material is essential for the diagnosis, staging, and molecular testing. For patients with advanced stage lung cancer the main method to acquire tumor material is through endobronchial ultrasound-guided transbronchial needle aspiration (EBUS-TBNA) [3]. Needle aspirates from EBUS-TBNA are normally processed into formalin fixed paraffin-embedded (FFPE) tissue blocks, which are then used for the sequential formal tissue diagnosis and molecular/genomic testing. There are clinical challenges with the increasing requirement for more molecular diagnostic tests and the small biopsy specimens available for analysis. There is a need to evaluate methods that could fast track more comprehensive genomic testing to improve the care of patients with advanced lung cancer.

For non-small cell lung cancer (NSCLC) the current clinical guidelines recommend testing for genomic alterations in *EGFR*, *BRAF*, *ROS1*, *ALK*, *KRAS*, *NTRKs*, *MET*, *RET*, and *ERBB2* genes [4,5]. Receiving molecular testing for these genes before first-line therapy influences patient clinical outcomes [6,7]. Even with the recognition of how valuable genomic testing is for treatment selection for advanced NSCLC [4,8], comprehensive genomic testing is not widely adopted in the clinic [5,9]. A recent report [4] of 3050 advanced non-squamous NSCLC patients showed that only 18.8% received comprehensive genomic testing, and 25% of patients did not receive genomic testing before starting treatment. Another study evaluating 38,068 patients newly diagnosed with advanced NSCLC [8] suggested that 49.7% of patients did not receive adequate testing for all actionable mutations. Reasons why patients missed out on complete genomic testing included the following: the biopsy was too small or contained insufficient tumor cells or testing took too long, lacked sensitivity, or did not cover the full repertoire of genes. In a reanalysis of data from 47,271 solid tumors sequenced between 2017 and 2022, the authors supported the importance of comprehensive genomic testing. They reported an increase in tumors harboring tier 1 or tier 2 actionable mutations from 8.9% to 31.6% due to the expansion of newly approved biomarkers and therapeutics over the period of 5 years [10].

Despite the increasing number of biomarkers for treatment, there is still a clinical need for new biomarkers that more effectively predict whether patients will respond to targeted or immunotherapy. The US Food and Drug Administration (FDA) approved immunotherapy for unresectable or metastatic solid tumors with PD-L1 expression [11], mismatch repair deficiency (dMMR) or microsatellite instability (MSI) [12], or high tumor mutational burden (TMB) [13]. The frequency of these biomarkers in lung cancer has recently been reviewed in a pan-tumor meta-analysis, finding dMMR in 1.6% of cases and high TMB in 27.5% of cases [14]. However, studies are emerging suggesting that other genomic features, such as mutations in *EGFR*, *KRAS*, *TP53*, *KEAP1*, and other genes, may impact the response to immunotherapy [15,16,17]. To fully characterize these and identify new treatment and more effective predictive biomarkers of response, large patient cohort studies with comprehensive genomic data and clinical follow-up of treatment response are needed. EBUS-TBNA specimens could represent a valuable source of tissue from advanced lung cancers to conduct such studies.

A limited number of studies with a small selected number of cases have explored EBUS-TBNA aspirates from advanced NSCLC for comprehensive panel sequencing [18,19] or whole exome (WES) [17] or whole genome sequencing (WGS) [20,21]. A pan-cancer study [22] compared TruSight Oncology 500 (TSO500) panel sequencing with WGS, including 22 FFPE lung cancer samples, and demonstrated similar abilities of TSO500 and WGS to detect SNVs, indels, copy number changes, and gene fusions. However, a large study evaluating the suitability of fresh EBUS-TBNA aspirates for comprehensive sequencing platforms has not been undertaken.

Here we recruited 220 patients with advanced NSCLC who underwent diagnostic EBUS-TBNA. We evaluated the DNA and RNA yield and tumor content of freshly collected cytology samples to determine the suitability of EBUS-TBNA aspirates for comprehensive genomic sequencing. Using the same nucleic acid extraction, for a subset of cases, we performed a cross-platform comparison between WGS, WES, and comprehensive panel sequencing (TSO500) evaluating the impact of the sequencing approach and read depth to detect genomic biomarkers that inform treatment decisions and for the discovery of potential novel biomarkers of treatment response. We also compared actionable mutations detected by fresh aspirate samples and FFPE material used in the standard of care testing collected during the same EBUS-TBNA procedure.

## 2. Materials and Methods

### 2.1. Cohort and EBUS-TBNA Specimen Collection

Recruited patients were referred, across seven major hospitals in Australia, for the investigation of mediastinal or hilar lymph nodes or masses with a high pre-test likelihood of malignancy and underwent EBUS-TBNA. EBUS-TBNA procedures were performed under either general anesthesia or conscious sedation. The most obviously involved node was sampled with up to five needle passes. Rapid on-site examination (ROSE) was performed by a pathologist using cytology smears (Diff-Quik) for malignant cell identification. The needle aspirate was divided between a standard of care (SOC) saline pot and a research pot. The division of the needle aspirates between the two pots occurred before the pathologist’s notification that the ROSE was positive. After the ROSE was declared positive, all aspirate material was collected for SOC. Research samples were collected simultaneously with half of each aspirate from 2 to 4 needle passes being collected either fresh frozen (at lead site) or in RNALater (for transporting to the centralized specimen processing lab, with 2–3 days). The SOC pot was processed into FFPE cell blocks for histopathology and biomarker testing, including immunohistochemistry for PDL1, ALK, and ROS1 (and fluorescence in situ hybridization if immunohistochemistry was equivocal or positive); single-gene *EGFR* testing; or small gene panel sequencing. Blood samples were collected in 4 mL EDTA tubes for genomic DNA extraction during the preparation for the EBUS-TBNA procedure.

This study was approved by the human research ethics committee of the lead site, Royal Brisbane & Women’s Hospital (HREC/17/QRBW/301 (ERM 35887)), and ratified by other hospitals and research organizations (University of Queensland—2018/HE001615; QIMR Berghofer—P2404). Patients were recruited between May 2018 and May 2022, consenting to the collection of blood and EBUS-TBNA samples for research purposes. No research results were returned to patients.

### 2.2. DNA and RNA Extraction from Research-Based EBUS-TBNA and Blood Specimens

EBUS-TBNA needle aspirates were homogenized using 1.4 mm ceramic beads (Sapphire Biosciences, Sydney, Australia) in a Precellys 24 device (Bertin Technologies, Montigny-le-Bretonneux, France). DNA and RNA were extracted using AllPrep^®^ DNA/RNA (Qiagen, Clayton, Australia). DNA was extracted from whole blood using the QIAamp^®^ DNA Blood Kit (Qiagen). DNA samples were quantified using the Qubit^®^ dsDNA broad-range Assay (Thermo Fisher Scientific Australia, Scoresby, Australia), and the DNA and RNA integrities of the tumor samples were evaluated using the 4200 TapeStation System (Agilent Technologies Australia, Mulgrave, Australia).

### 2.3. Single-Nucleotide Polymorphism (SNP) Arrays

Where sufficient tumor DNA was available (>250 ng), tumor and matched normal blood DNA were assayed using Illumina Global Screening Array-24 (Illumina Australia, Melbourne, Australia). The array data were used to calculate the tumor content in the samples using the qPure tool [23].

### 2.4. Sequencing

A subset of 40 patients were subjected to WES, WGS, and TSO500 panel sequencing. For WES and WGS, 1000 ng of DNA from each tumor and matched normal (blood) were sent to commercial sequencing providers (Appendix A). For WES, coding regions were captured using Sure Select Human all Exons Version 6 (Agilent) and sequenced in a NovaSeq 6000 (Illumina). For WGS, libraries were prepared using TruSeq DNA PCR Free, 150 bp paired end, sequenced in a NovaSeq 6000, or a PCR library was prepared as DNA nanoballs and sequenced in a DNBSEQ-G400 instrument at Beijing Genomics Institute (BGI). For TSO500 (Illumina), only tumor samples were sequenced, and 100 ng of tumor DNA and RNA were sequenced on a NextSeq 550 system with eight cases per run. The sequenced tumor samples had a tumor content ≥30% and >2500 ng of DNA to allow for all three sequencing approaches. The same DNA eluate was used for all sequencing platforms. A further six cases with low tumor content (<30%) were sequenced using TSO500 to assess panel sensitivity.

Appendix A shows sequencing metrics: WES had an average read depth of 126x (range 99–233) for normal blood and 222× (range 124–454) for tumors, and WGS had an average read depth of 36× (range 21–96) for normal blood and 67× for tumors (range 45–96). For TSO500, the tumor median exon read depth was 1361× (range 762–1734) with an average of 99.3 percent of exon bases covered by at least 50 reads. The RNA component of the TSO500 had on average 23,315,012 on-target reads per sample (range 14,702,314–28,192,412), one sample failed RNA minimal read counts (9 million), and three samples had no quantifiable RNA.

### 2.5. Mutation Detection

WES and WGS data sets were adapter trimmed using Cutadapt (version 1.9) and aligned to the GRCh37 assembly using BWA-MEM (version 0.7.12) and SAMtools (version 1.1). Duplicate reads were marked with Picard MarkDuplicates (https://broadinstitute.github.io/picard, accessed on 8 September 2023 (version 1.129); http://picard.sourceforge.net, accessed on 8 September 2023). The average read depth was estimated using qCoverage (version 0.7pre; https://github.com/AdamaJava/adamajava, accessed on 8 September 2023). Somatic SNVs and indels were detected using an established pipeline [24,25,26], where a dual calling strategy was used to detect SNVs, with the consensus of two different tools being used for downstream analysis: qSNP (version 2.0) [27] and GATK HaplotypeCaller (version 3.3-0) [28]. The detection of indels (1–50 bp) was carried out using GATK. Variant annotation for gene consequence was performed using SnpEff [29]. Mutations were called if at least 4 reads harbored the alternative allele in positions with a minimum 8 read depth in the tumor sample and 12 reads in the matched normal sample. The minimum allele frequency of 5% was used to report mutations. Structural variants were determined on WGS as previously described [24,25]. Only structural rearrangements with predicted loss of function or gene fusion were reported. The copy number and genome ploidy for WGS were determined using ascatNGS [30] and for WES using Sequenza [31]. Copy number segments were annotated against Ensembl genes (version 75). Genes were considered amplified if the copy number was >5 when the average genome ploidy was ≤2.7 or ≥9 if the average genome ploidy was >2.7. For the analysis of driver mutations, we considered the loss of heterozygosity (LOH) status of genes to inform potential biallelic inactivation.

TSO500 mapping and mutation analysis was performed using TruSight Oncology 500 Local App Version 2 (Illumina) for tumors only. All samples passed vendor quality control criteria: median insert size≥ 70 bp, median exon coverage ≥150×, and percentage of exons with coverage of at least 50× ≥ 90%. Mutations were identified as somatic using vendor filtering based on public databases (excludes any variant with an observed allele count ≥10 in any of the GnomAD, exome, genome, and 1000 genomes databases). Only somatic mutations with allele frequency ≥5% were reported. Copy number changes were reported in the form of fold change on the normalized read depth of the tested sample relative to the normalized read depth in diploid genomes. For TSO500, we used a fold change of 2, which in samples with 30% tumor content was estimated to represent ~9 copies of the genes reported as per vendor indication.

### 2.6. Clinically Actionable Mutations

OncoKB-MSK’s Precision Oncology Knowledge Base (https://www.oncokb.org, accessed on 8 September 2023) was used to identify actionable mutations with clinical evidence. Mutations were manually reviewed using the Integrative Genomics Viewer (IGV) [32] and were annotated with therapeutic levels of evidence from OncoKB (consulted on 8 September 2023) as follows: tier 1, an FDA-recognized biomarker predictive of response to an FDA-approved drug; tier 2, a standard of care biomarker recommended by professional guidelines as predictive of response to an FDA-approved drug; tier 3, compelling clinical evidence or investigational biomarker of response; tier 4, biological evidence (hypothetical) biomarker of response; R1, standard of care biomarker of resistance to an FDA-approved drug; or R2, compelling clinical evidence of being predictive of resistance to a drug. Mutations without therapeutic evidence were labeled as somatic mutations.

### 2.7. Tumor Mutation Burden (TMB), Microsatellite Instability (MSI), and Other Potential Biomarkers of Immunotherapy

For TSO500, TMB and MSI were estimated using TruSight Oncology 500 Local App Version 2 (Illumina) as per the vendor description. For WES and WGS, TMB was calculated by dividing the number of non-synonymous somatic mutations by the approximate size of the protein coding region of the genome (30 Mb). A TMB threshold of 10 mutations/Mb was used to define samples with high versus low TMB; this is a common threshold used in previous studies [17,33,34] and FDA approved for treatment of metastatic unresectable solid tumors when using FoundationOneCDx assay. MSI was estimated for WGS and WES using msisensor v0.2 [35] with paired tumor-normal data. A sample was considered microsatellite unstable if the percentage of sites exhibiting microsatellite instability in WGS or WES was ≥3.5% [35] and for TSO500 if >10% [22].

### 2.8. Mutational Signatures

SigProfilerAssignment version 0.030 (https://github.com/AlexandrovLab/SigProfilerAssignment, accessed on 8 September 2023) was used to assign the somatic mutations detected by each platform for individual samples to previously known mutational signatures using cosmic v3.3 signatures as the reference signature matrix. For exome and TSO500 platform signature assignment, the option “Exome = true” was used. The contributions of detected signatures identified in each sample with the same proposed etiology were combined, i.e., Tobacco (SBS4, 29 and 92), clock-like/age (SBS1 and 5), and APOBEC (SBS2 and 13). Other signatures that were less frequent or were with an unknown etiology were grouped together as “other”. The combined contribution of these mutational signatures was compared between sequencing platforms.

## 3. Results

### 3.1. Cohort

The cohort consisted of 220 patients diagnosed with advanced NSCLC by EBUS-TBNA (Table 1). Most patients were male (60%) and had smoking history (81.4%) and stage III or IV disease (89%). Cases were classified as NSCLC not otherwise specified (n = 63), adenocarcinoma (n = 98), or squamous cell carcinoma (n = 59).

#### Results of SOC Testing

Of 161 non-squamous NSCLC cases, 31 (19.3%) had insufficient tissue for testing, and 7 (4.3%) were not tested, resulting in 23.6% of cases not receiving testing to guide treatment (Table 1). Of those that received SOC genomic testing, 46 cases (37%) underwent only single-gene (*EGFR*) PCR testing, and 74 (58%) received panel sequencing. Mutations were detected in 41 cases (33% of the 123 cases that underwent SOC testing), 19 with *KRAS* mutations, 14 *EGFR*, 5 *BRAF*,1 *ROS1*, 1*ALK*, and 1 *ERBB2*; only 4 out of 59 cases diagnosed as squamous cell carcinoma received testing, with no mutations reported.

### 3.2. DNA and RNA Yield from EBUS-TBNA Aspirates from 220 NSCLC Cases

We applied thresholds for DNA yield and tumor content to determine the proportion of samples that could undergo sequencing on different platforms. The threshold for TSO500 or WES was ≥100 ng DNA, and for WGS it was ≥1000 ng of DNA with ≥30% tumor content. Sixteen samples (7.3%) were deemed “insufficient”, i.e., no material for extraction (n = 7) or <100 ng of DNA yield (n = 9) (Figure 1a). A total of 204 samples (92.7%) had ≥100 ng of DNA, allowing for a comprehensive panel and potentially WES; 32 of those samples had between 100 ng and <1000 ng, and the remaining 172 had ≥1000 ng of DNA. A total of 192 samples had sufficient DNA for SNP arrays to estimate the tumor content (Appendix A), and 109 samples (49.5%) qualified for WGS (Figure 1b). High DNA yield was not associated with high tumor content (*p* = 0.809, Fisher’s exact test), with 63 samples yielding ≥1000 ng of DNA having <30% tumor content (Figure 1a).

Actionable gene fusions can be detected using WGS or through the RNA sequencing component of TSO500 [22]. In total, 183 (83%) cases had sufficient RNA (>100 ng) for panel testing (Appendix A); for samples with a DNA yield between 100 ng and 1000 ng, 65% had >100 ng of RNA, whereas in samples with >1000 ng of DNA, 94% had >100 ng of RNA (Figure 1a), suggesting as expected that the yield of DNA and RNA were not independent (*p* < 0.0001, Fisher’s exact test). Out of 109 samples suitable for WGS, 84.4% had >500 ng of RNA that could be suitable for RNASeq from EBUS-TBNA aspirates (Appendix A).

### 3.3. Comparing TSO500, WES, and WGS Sequencing Platforms

For 40 cases (29 non-squamous NSCLC and 11 squamous cell carcinomas) the specimen had sufficient DNA (>2500 ng) and tumor content >30% to enable a cross-platform comparison. For these samples, DNA from the same extraction was subjected to WGS, WES, and TSO500 (Appendix A). The detected mutations were also compared with SOC testing performed using DNA extracted from FFPE diagnostic samples collected during the same EBUS-TBNA procedure. Tier 1 mutations were clinically reported for seven of the 40 cases (six *EGFR* and a *KRAS*: p.Gly12Cys), all of which were also detected by each comprehensive sequencing platform in the fresh sample (Figure 2). An additional three cases harbored tier 1 mutations detected by all three sequencing platforms (Figure 2), which were not reported by SOC (case A, *MET*: c.2942-29_2942-2del—exon 14 skipping-splice region; case I, *KRAS*: p.Gly12Cys; and case C, *ERBB2*: p.Ala771_Tyr772insAlaTyrValMetAla—exon 20 insertion). Two of these cases (A and I) only received *EGFR* (PCR) testing due to insufficient FFPE material for panel testing. Note that case C received SOC panel testing in 2019; however, *ERBB2* mutations were not an approved actionable target until 2022 [10]. We reviewed the sequencing data of the clinical testing, and the mutation was detected in the FFPE material but was not reportable as actionable at the time of the clinical testing.

Tier 2 or 3 mutations (*MET* and *FGFR1* amplifications) were detected by at least two platforms in six additional cases (15%; Figure 2). They were not detected in SOC testing due to the use of single-gene (*EGFR* only) testing or small panel sequencing, which did not include evaluation of copy number alterations. Case G harbored both a tier 1 *EGFR* mutation (p.Glu746_Ala750del) and compelling evidence of resistance (R2) mutation *EGFR*: p.Asp761Tyr. Both mutations were detected by all comprehensive platforms and SOC (Figure 2, Appendix A). Tier 4 mutations were identified in a further 10 cases (25%), including two reported by SOC testing (*BRAF* and *KRAS*; Figure 2, Appendix A). Tier 4 mutations are not yet suitable to inform clinical decisions for treatment. However, these results show the potential of comprehensive sequencing to detect mutations that might emerge as future treatment options or could facilitate the enrollment of patients in clinical trials. In total, 14 cases (34%; one adenocarcinoma, five NSCLC, and eight squamous cell carcinomas) had no potentially actionable mutations detected.

### 3.4. Discrepancies of Actionable Mutations Detected between Sequencing Platforms

Of the 37 mutations detected that can potentially affect treatment response (tiers 1–4 and R2), 28 were detected by all three platforms (Figure 3a), five mutations were detected by two platforms, and four were detected by only one platform. Since all sequencing was performed using the same DNA extraction, the differences in mutation detection between platforms will be platform or analysis pipeline specific rather than being related to intra-tumor heterogeneity. All 10 tier 1 mutations were detected by all three sequencing platforms. Discrepancies in mutation detection across platforms occurred for mutations with lower levels of evidence of predictive response (Figure 3b). IGV review of the nine mutations not detected by one or two platforms (Appendix A) could be attributed to a low read depth at a specific position in a particular platform, a low number of reads harboring the mutation (tumor content), the position not being captured by a particular platform, or differences in the analysis pipelines used (thresholds and filters). For example, TSO500 detected a tier 4 mutation in *ARID1A* (p.Asp1850Glyfs—case P, Figure 2), which on manual review of the data was present in WGS/WES but was filtered out in the respective analysis pipeline due to the mutation being located next to a homopolymer region. Tier 2 *MET* amplification was detected in both WGS and WES but not TSO500 in case R, which had a 30% tumor content. For this case, the copy number just passed the threshold set for amplification in WGS and WES but for TSO500 was reported as FC = 1.891, which represents approximately eight copies of the gene just below our threshold (FC ≥ 2, approximately nine copies; Appendix A). The number of mutation types (SNV, indel, and amplification) detected by each platform was similar (Figure 3c), suggesting that the platforms were not adversely affected by or did not favor specific mutation types.

### 3.5. TSO500 on Low-Tumor-Content Samples

From 204 cases with ≥100 ng DNA, 71 (34.8%) had <30% tumor content (Appendix A, Figure 1a). We sequenced a further seven cases (six non-squamous NSCLC and one squamous cell carcinoma) with tumor content <30% (Appendix A) using TSO500 only. Here TSO500 detected a *ROS1* fusion and *EGFR*: p.Leu861Gln in two cases, with 12% and 29% tumor content, respectively. These mutations were reported in SOC. TSO500 identified an additional tier 1 *RET* (Exon 12)–*KIF5B* (Exon 15) fusion, a tier 2 *MET* amplification, and two tier 4 mutations (*PTEN* and *STK11*). These results support the potential of high-depth comprehensive panel sequencing for samples of low tumor content.

### 3.6. Detection of Potential Biomarkers of Immunotherapy Response

Programmed death-ligand 1 (PD-L1) staining [11], the tumor mutation burden (TMB) [13], and microsatellite instability (MSI) [12] are approved biomarkers of immunotherapy response. Here we evaluated TMB and MSI estimations across the three sequencing platforms for 40 cases (Appendix A). Only cases U and AD had MSI estimates higher than a threshold used for other cancer types [35] of 3.5% of sites unstable genome wide. For case U, WGS and WES had MSI scores above the threshold, but this case did not have a high burden of indels or SNVs, which were previously associated with high MSI [36]; somatic mutations were detected in *MSH2* and *MSH6* genes, but there was no loss of heterozygosity that could suggest biallelic inactivation of those genes and support potential MSI. For case AD, only WGS passed the threshold (>3.5%) MSI, and this case had a high TMB and high indel burden but no mutations in the mismatch repair genes. No mutational signatures associated with DNA mismatch repair and/or microsatellite instability were observed in WGS or WES sequencing data for these cases (Appendix A). For TSO500 all samples had MSI in <10% of sites tested. Taken together, these results did not offer strong support for microsatellite instability in this cohort and showed that by using comprehensive sequencing other features of the genome could be considered to support MSI.

The TMB status (Figure 4a, Appendix A) was consistently classified, according to the threshold of 10 mutations/Mb, as either high or low TMB across all three platforms for 31 samples (78%). TMB was slightly lower in WES compared with WGS, but the correlation between these technologies was 0.99 (Figure 4b; Appendix A). The TSO500 TMB estimates were higher compared with WES in 29 cases (73%), in agreement with previous reports [37,38], with a correlation of 0.92 (Figure 4c). The correlation between WGS and TSO500 was 0.94 (Figure 4d). Importantly, TMB was discrepant between platforms in nine cases; in five of those cases the WES TMB estimates were <10 mutations/Mb and discordant to the other two platforms, and in five cases the tumor content was <50% (Figure 4a, Appendix A), suggesting that a lower tumor content could impact the detections of mutations and contribute to the disparity in TMB estimates between platforms.

Recent studies suggest that other genomic features may play a role in predicting the immunotherapy response, including the indel burden; mutational signatures [17]; and mutations in genes such *EGFR*, *KRAS*, *STK11*, *KEAP1, TP53, SMARCA4, ATM,* and *TERT* [17,39,40,41]. We evaluated if the above genomic features could be similarly detected by WGS, WES, and TSO500 (Appendix A, Appendix A). Despite our small cohort, as expected, these genes were frequently mutated in non-squamous NSCLC but not in the squamous subtype, highlighting the importance of future studies evaluating other lung cancer subtypes and that WGS of EBUS-TBNA could facilitate those discovery studies. For instance, 23% and 33% of non-squamous NSCLC had point mutations in *KRAS* and *EGFR*, respectively, consistently detected by all three platforms, except in case AB, which had 32% tumor content. Here the *KRAS* mutation was not detected by WES where fewer than four reads harbored the mutation, below the minimum evidence required to confidently report mutations within the analysis pipeline for WES and WGS (Appendix A).

As expected, WGS allows the detection of structural rearrangements in genes across the genome, which is not possible by WES or TSO500 (which can detect gene fusions by RNA capture and sequencing but only for a small set of genes). This is especially important when evaluating genomic features involving genes such *KEAP1, STK11*, and *SMARCA4*, where 28%, 50%, and 40%, respectively, of the mutations affecting those genes were loss of function structural rearrangements (Appendix A).

### 3.7. Mutational Signatures in EBUS-TBNA Samples

As we used the same DNA for WGS, WES, and TSO500, the extraction of mutation signatures is expected to be impacted only by the number of mutations detected by each platform (Figure 5, Appendix A). Tobacco, clock-like/age, and APOBEC were the most frequent signatures across the cohort in WGS and WES sequencing. The number of mutations assigned to tobacco and APOBEC signatures was highly correlated between WGS and WES (r ≥ 0.9, *p* < 0.001; Figure 5b,c), with the age-related signature showing only a moderate positive correlation (r = 0.5, *p* = 0.001; Figure 5d). TSO500 assays a small portion of the genome, resulting in the detection of a lower number of somatic mutations (Appendix A), which is expected to impact the extraction of meaningful mutational signatures. This clearly impacted meaningful signature extractions (Appendix A), yielding low sample-specific measures (cosine similarity and correlation) for how well the mutational profile of a sample could be explained by the COSMIC mutational signatures (Appendix A). Consequently, the correlation of the number of mutations assigned to signatures between TSO500 and WGS was low (r ≤ 0.45; Figure 5e–g). There were two highly mutated samples (patients O and AD with >30 mutations/Mb); their removal did not affect the correlations of signatures detected in WGS and WES but further negatively impacted the correlations of signatures estimated by TSO500 versus WGS (Appendix A). These results suggest, as expected, that panel sequencing with low numbers of detected mutations are not suitable for mutational signature estimations.

## 4. Discussion

Genomic testing for lung cancer patients is a critical component of informed treatment decision making. However, current clinical testing rates need to be improved. Similar to the published literature [4,5], approximately a quarter of patients in our cohort of 161 non-squamous NSCLC did not receive any genomic testing, and 40% of tested cases only received single-gene (*EGFR*) testing. There are various reasons for the lack of or for incomplete genomic testing in the clinic [8], with insufficient diagnostic tissue and the lack of comprehensive testing platforms being important contributors.

Here we evaluated the largest EBUS-TBNA specimen cohort to see whether the simultaneous collection of FFPE material for pathology-based diagnostics and fresh material (frozen or in RNAlater) for genomics could address the clinical need to improve genomic testing rates for lung cancer patients. Importantly, our study demonstrates that the additional collection of fresh aspirate tissue did not adversely affect the SOC clinical testing rates compared with previous reports, while also showing that comprehensive genomic testing could be possible for most patients. From 220 freshly collected EBUS-TBNA specimens, there was sufficient DNA in >90% of cases to enable comprehensive panel sequencing for currently known actionable point mutations and gene amplifications, and there was sufficient RNA (>100 ng) in >83% of specimens to enable testing of gene fusions. This suggests that fresh EBUS-TBNA aspirates could be a valuable alternative source of tumor material for genomic testing. The limitation here is that tumor content cannot be quickly estimated in fresh material by microscopic inspection, but our data suggest that even with a low tumor content of <30%, a comprehensive panel can detect actionable mutations.

The specimen type and availability are critical to the success of genomic testing. As EBUS-TBNA provides the only diagnostic tissue for most patients with advanced lung cancers, a wider adoption of freshly collected EBUS-TBNA aspirates and comprehensive panel sequencing could improve the detection of actionable mutations and outcomes for these patients. The College of American Pathologists, the International Association for the Study of Lung Cancer, and the Association for Molecular Pathology indicate that any cytological sample with well-preserved material can be used for molecular testing [42]. Marrying the need to enhance testing capability at the same time as obtaining a diminishing tumor sample volume (by biopsy or aspirate) is challenging. Nevertheless, we previously showed the clinical value of cytology slides, such as Diff-Quik stained slides for comprehensive panel sequencing [18], and here we further demonstrate the broad utility of freshly collected specimens. Other important technological advances are enabling fast-tracked testing using freshly collected aspirates [43,44]. We therefore advocate the use of alternative sources of tumor tissue (fresh and/or Diff-Quik) to expedite comprehensive genomic testing and at the same time spare the FFPE tissue block for other diagnostic tests.

In a multi-platform comparison, we showed that TSO500, WES, and WGS detected all tier 1 actionable mutations reported by SOC testing using the FFPE specimen collected in the same EBUS-TBNA procedure. Comprehensive platforms detected additional tier 1 mutations in 7% of patients that were not clinically tested or reported due to insufficient tissue, incomplete testing, or the mutation not being clinically actionable according to society guidelines at the time of testing [10]. Further, in specimens with low DNA yield and <30% tumor content sequenced by TSO500, tier 1 fusion genes were detected in *ROS1* (also reported by SOC) and *RET*. Our findings reinforce the immediate clinical value of comprehensive panel sequencing for actionable mutation detection and its applicability for most patient samples, even those with low tumor content. The adoption of comprehensive panels has been shown to have diagnostic and economic value when compared with serial single-gene testing in advanced NSCLC [2,45,46,47]. Overall, it is previously reported that multigene panels increase the proportion of patients with access to targeted therapy when compared with combinations of single-gene tests [46], resulting in an increase in life years or quality-adjusted life years [2,47] while being cost-neutral or cost-saving to the health system [2,45,46,47]. Previous studies evaluated the cost-benefit associated with the implementation of comprehensive panels using FFPE specimens. This does not address a current clinical limitation of insufficient material remaining for genomic testing after other diagnostic tests are performed. Unfortunately, our study is underpowered for a cost-benefit analysis. However, there is an urgent need to evaluate the cost-benefit of introducing alternative sources of tumor tissue and different types of genomic testing in the clinical setting to improve the care of patients with advanced lung cancer.

Our study also evaluated approved biomarkers of immunotherapy response, microsatellite instability (MSI) and TMB. There were no high MSI cases, and TMB was highly correlated across the three sequencing platforms. TSO500 TMB was frequently higher than WES TMB, as reported by others [37,38], likely due to the genomic regions sequenced in panels being smaller and targeting genes recurrently mutated in cancers. In our study, discordant TMB statuses (above or below of the threshold of 10 mutations/Mb) were more frequent in WES versus other platforms and in low-tumor-content samples, suggesting that the lower tumor content could contribute to the disparity of TMB statuses between sequencing platforms. This trend is supported by a previous study [48] that showed that panels have greater sensitivity for the detection of mutations in low-tumor-content samples due to greater sequencing depth compared with WES. This has enabled panel TMB to more accurately predict progression-free survival after immunotherapy [48].

Another aspect of our study was to evaluate a large cohort of EBUS-TBNA specimens for WGS suitability, opening the possibility to identify novel biomarkers of treatment response in new discovery studies. The power of WGS in precision oncology is gaining traction in other settings [49,50,51], including in selected EBUS-TBNA specimens [20,21]. Here we showed that WGS is possible in ~50% of EBUS-TBNA aspirates (sufficient tissue/DNA and tumor content). These numbers could be even higher as we used a stringent threshold of 1000 ng of DNA for WGS. Further advances in sequencing technology will, in time, improve the specific sample requirements for WGS and other clinical challenges of implementation [52]. In the short term, WGS, together with EBUS-TBNA specimens, perhaps offers the most comprehensive genomics for the potential discovery of new biomarkers of response or resistance to targeted therapies and immunotherapy. The recent publication from Genomics England [51] exemplifies the potential of WGS in oncology, but in their study the majority of lung specimens were surgery resections; EBUS-TBNA could facilitate access to tumor tissue of advanced lung cancer cases.

There is a need for more robust predictive biomarkers of immunotherapy response as currently, only approximately 20 to 30% of NSCLC patients have clinical benefit from the treatment [53]. Recent studies have indicated that certain genomic features [39,40,41,54] or co-mutations [41] in the tumor genome may help to stratify patients that will benefit or not from immunotherapy. WGS has the advantage of detecting structural variants that can contribute to the disruption of genes impacting their function in the cancer genome. Mutational signature burdens (age, tobacco, and APOBEC) have been recently associated with the response to checkpoint blockade [17]. In our comparison across sequencing platforms, the proportions of the mutations assigned to mutational signatures across samples were highly correlated between WGS and WES, but as expected, the confidence of the extracted mutational signatures from panel sequencing (a low number of mutations) was poor. WGS is best placed for discovery studies as it allows for the detection of all types of mutations and other genomic features such signatures in a single test. EBUS-TBNA is the main source of tissue in metastatic lung cancers where immunotherapy is currently an approved treatment. The WGS/EBUS-TBNA combination has substantial discovery potential for treatment response biomarkers in cohorts with clinical follow-up.

## 5. Conclusions

Our study demonstrated the enormous potential of fresh EBUS-TBNA aspirates for comprehensive sequencing. We showed that fresh EBUS-TBNA aspirates were suitable for three comprehensive sequencing platforms with comparable detections of actionable mutations in samples with ≥30% tumor content. Comprehensive panels hold advantages in the clinical workflow where specimens with low DNA yield and tumor content are common. However, we showed the feasibility of WGS in EBUS-TBNA aspirates and their discovery potential. There is a clinical need for better characterization/identification of biomarkers of response to treatment in large cohort studies with comprehensive genomics. Given that EBUS-TBNA is widely used in the diagnosis of advanced lung cancers, samples from these procedures can therefore fit the requirement for worldwide cohort studies using comprehensive genomics of large cohorts with clinical follow-up that could radically improve prospects for lung cancer patients.

## Figures and Tables

**Figure 1 cancers-16-00785-f001:**
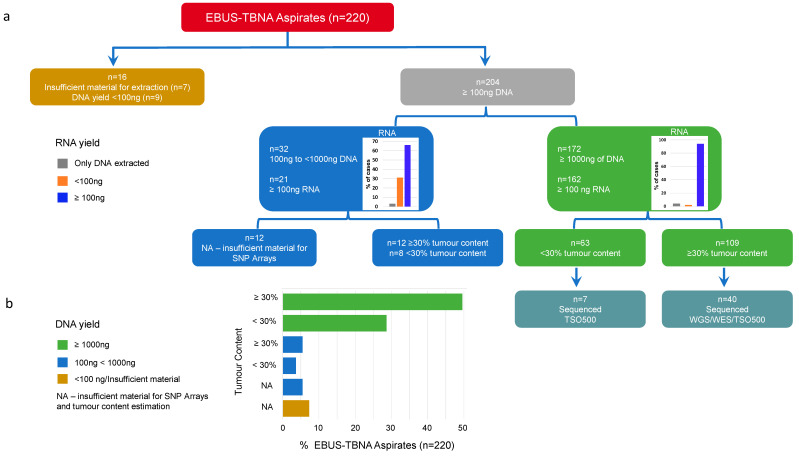
Summary of the cohort of 220 EBUS-TBNA fresh samples from advanced non-small lung cancers. (**a**) Diagram with number of samples grouped by DNA yield and those with single-nucleotide polymorphism (SNP) genotyping array estimation of tumor content. The number of samples sequenced is shown at the end of the diagram. (**b**) Percentage of EBUS-TBNA aspirates (*x*-axis) and tumor content (*y*-axis), colored by the DNA yield. NA = insufficient material for SNP arrays.

**Figure 2 cancers-16-00785-f002:**
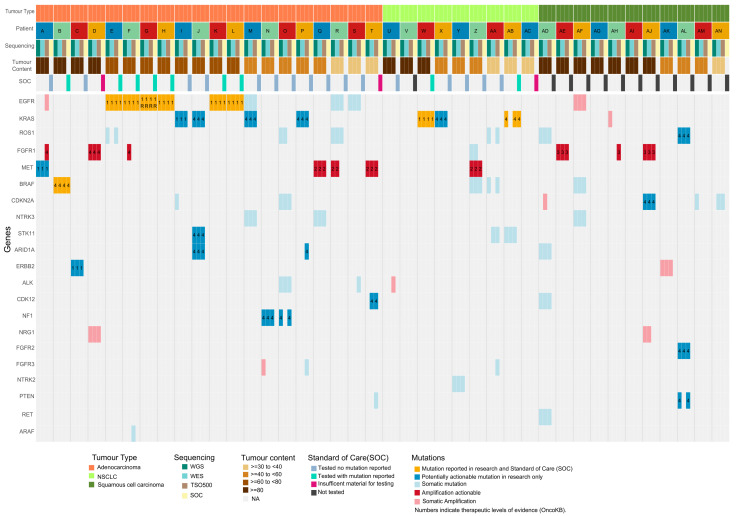
Oncoplot of mutations detected in 40 patients across WGS, WES, TSO500, and standard of care (SOC) in genes reported to harbor potential actionable mutations for NSCLC. Patients are indicated by letters, and each column reports results for a sequencing platform and SOC. Patients are ordered by tumor type followed by percentage of tumor content. Actionable genes were selected from evidence reported on OncoKB MSK’s Precision Oncology Knowledge Base (consulted on 8 September 2023). Genes were included if a mutation was detected in at least one tumor by one sequencing platform. The numbers reported in the plot indicate therapeutic levels of evidence for the mutation identified (1—Tier 1, FDA-recognized and approved drug; 2—Tier 2, standard of care by professional guidelines to an approved drug; 3—Tier 3, investigational, with clinical evidence; 4—Tier 4, hypothetical, biological evidence; R—Tier R2, compelling clinical evidence as being predictive of resistance). Note: Patient G had a Tier 1 mutation (1) and a R2 resistance mutation (R).

**Figure 3 cancers-16-00785-f003:**
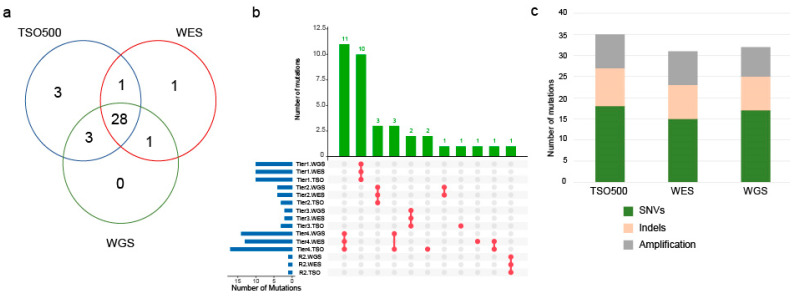
Number and types of mutations detected across the three sequencing platforms in 40 cases. (**a**) Venn diagram shows the overlap of mutations (tiers 1–4) detected by each sequencing platform. (**b**) Top panel bar chart shows numbers of mutations of a given Tier (1–4). Upset plot below shows the success of each platform in detecting those mutations. Red dots indicate which of the three platforms detected these mutations. For example, there were 11 Tier 4 mutations where all three platforms detected them; there was one Tier 2 mutation where only WES and WGS detected it. Grey dots represent no value. (**c**) Number of mutations (tiers 1–4) detected by each platform colored by mutation type (SNV, indels, and amplification).

**Figure 4 cancers-16-00785-f004:**
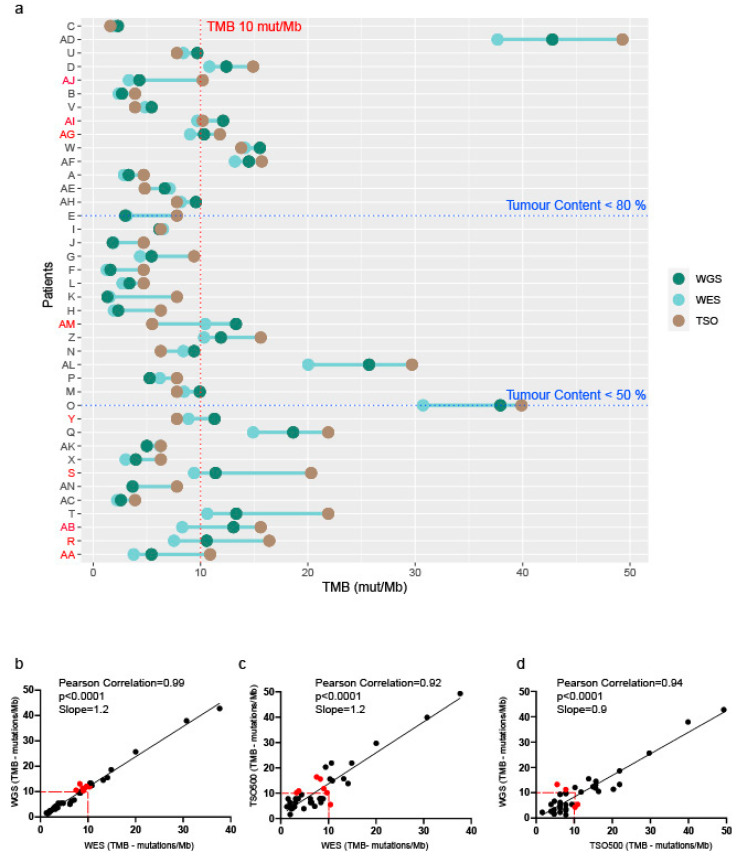
Tumor mutation burden (TMB) estimated from each sequencing platform. (**a**) TMB values for each patient and each sequencing platform (WGS, WES, and TSO500 panel). The TMB threshold of 10 mutations/Mb used to determine the TMB status (high or low) is shown with the red dotted line. Patients are ordered by tumor content. Tumor content thresholds of 80% and 50% are shown with blue dotted lines. Patients with discordant TMB status between platforms are in red. (**b**) Pearson correlation of TMB estimated from WGS and WES (n = 40). (**c**) Pearson correlation of TMB estimated by WES and TSO500 (n = 40). (**d**) Pearson correlation of TMB estimated by WGS and TSO500 (n = 40). Red dots indicate samples with TMB discordance (considered high or low TMB using the threshold of 10 mutations/Mb) between platforms.

**Figure 5 cancers-16-00785-f005:**
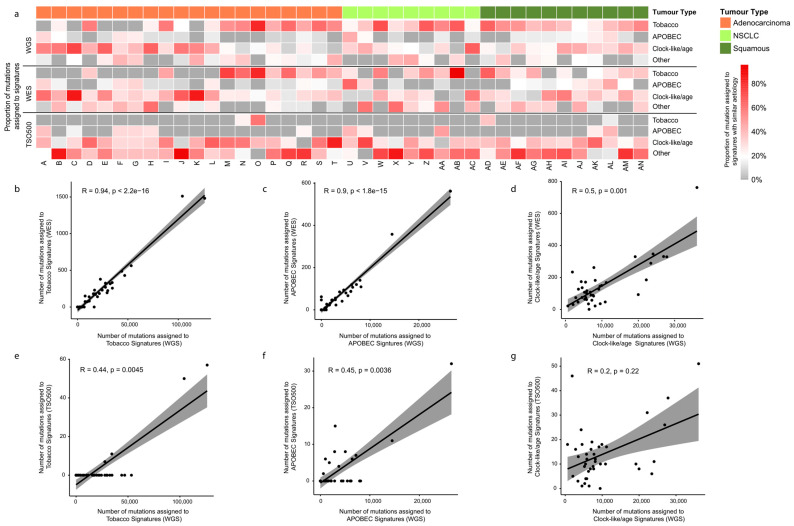
Cross-platform comparison between prominent mutational signatures extracted from EBUS-TBNA sequencing data. (**a**) To allow for visualization across platforms, the proportions of mutations assigned to mutational signatures with similar etiologies were combined, i.e., Tobacco (SBS4, 29, and 92), clock-like/age (SBS1 and 5), and APOBEC (SBS2 and 13). Here they are presented as a percentage of the total number of mutations detected in each sample for each sequencing platform. The proportions of mutations assigned to other signatures were combined together and are shown as “Other” (see Appendix A for the distribution of mutation signatures in these specimens). Spearman correlations across the 40-sample cohort between the number of mutations assigned to (**b**) tobacco signatures in WGS and WES data, (**c**) APOBEC signatures in WGS and WES data, (**d**) clock-like/age signatures in WGS and WES data, (**e**) tobacco signatures in WGS and TSO data, (**f**) APOBEC signatures in WGS and TSO data, and (**g**) clock-like/age signatures in WGS and TSO data.

**Table 1 cancers-16-00785-t001:** Demographic of the NSCLC cohort (n = 220 patients).

Characteristics (n = 220)	Number of Patients (%)
Age, average (range)	67.17 (42–88)
Sex (n = 220)	
Female	87(40%)
Male	133 (60%)
Histology (n = 220)	
Adenocarcinoma	98 (44.5%)
Non-small cell lung cancer	63 (28.6%)
Squamous cell carcinoma	59 (26.8%)
Tumor stage (n = 220)	
IA	3
IB	7
IIB	9
IIIA	43
IIIB	41
IIIC	17
IVA	40
IVB	55
Information not available	5
Lymph node station/mass * (n = 220)	
2R	1
3	2
4R	45
4L	19
7	74
10	9
11R	36
11L	21
12	3
Tumor mass	8
Information not available	2
Smoking status (n = 220)	
Current smoker	67 (30.5%)
Ex-smoker	112 (50.9%)
Never smoked	17 (7.7%)
Information not available	24 (10.9%)
Standard of care testing (n = 161 non-squamous NSCLC)	
Single-gene test (EGFR)	46 (28.6%)
Small NGS panel	74 (45.9%)
Insufficient tissue for testing	31 (19.3%)
Not tested	7 (4.3%)
Information not available	3 (1.9%)
*ALK* and *ROS1* testing (n = 161 non-squamous NSCLC)	
Positive ROS1	1 (0.5%)
Positive ALK	1 (0.5%)
Equivocal	2 (1.2%)
Negative	112 (69.6%)
Not tested	14 (8.7%)
Insufficient tissue	26 (16.1%)
Information not available	5 (3.1%)
PD-L1 (n = 220)	
Tested	160 (72%)
Not tested	23 (10.5%)
Insufficient tissue	24 (10.9%)
Information not available	13 (5.9%)

* Lymph node station includes 38 cases where the mass was contiguous with the sampled lymph node (Appendix A).

## Data Availability

Sequencing data will be available in the European Genome-phenome Archive (EGA) under the accession code EGAS00001007708.

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
