# Peer review of "Evaluation of Endobronchial Ultrasound-Guided Transbronchial Needle Aspiration (EBUS-TBNA) Samples from Advanced Non-Small Cell Lung Cancer for Whole Genome, Whole Exome and Comprehensive Panel Sequencing"

_cancers, 2024, doi:10.3390/cancers16040785_

Round 1

Reviewer 1 Report

Comments and Suggestions for Authors

The manuscript submitted by Fielding and colleagues reported the feasibility of comprehensive genomic profiling of a large cohort of advanced NSCLC using EBUS-TBNA aspirates.

The study design and the data are very interesting, adequately presented and discussed, therefore the manuscript is suitable for publication in Cancers:

Minor observations:

Even though the authors already reported that the “most involved node” were sampled by EBUS-TBNA, it would be more complete to add the specific sampled sites to Table 1 patient data.

Many papers focused on the limitations of the EBUS-TBNA procedures and the frequent low diagnostic yield of cytological samples obtained. As the presented data strongly support the use of this procedure for a successful comprehensive molecular characterization of NSCLC, I would suggest to discuss in a more detailed way also the limitations of this procedure (if any in the authors experience): this could help to encourage a wider use of this approach.

Figure 1 is very useful as it recap the samples and how they were used in the different analysis: I would enlarge it (full page?) in order to make the study plan clearer.

Author Response

Thank you very much for taking the time to review our manuscript.

Reviewer 1:

The manuscript submitted by Fielding and colleagues reported the feasibility of comprehensive genomic profiling of a large cohort of advanced NSCLC using EBUS-TBNA aspirates.

The study design and the data are very interesting, adequately presented and discussed, therefore the manuscript is suitable for publication in Cancers:

Minor observations:

Even though the authors already reported that the “most involved node” were sampled by EBUS-TBNA, it would be more complete to add the specific sampled sites to Table 1 patient data.

Response: We have included the sampled sites in Table 1. And individual information was added to Table S2.

Many papers focused on the limitations of the EBUS-TBNA procedures and the frequent low diagnostic yield of cytological samples obtained. As the presented data strongly support the use of this procedure for a successful comprehensive molecular characterization of NSCLC, I would suggest to discuss in a more detailed way also the limitations of this procedure (if any in the authors experience): this could help to encourage a wider use of this approach.

Response: We have expanded the section in the discussion to include some of the limitations. See line 481.

Figure 1 is very useful as it recap the samples and how they were used in the different analysis: I would enlarge it (full page?) in order to make the study plan clearer.

Response: We have enlarged the figure (full page).  We also increase the sizes of Figures 3 and 4 but we hope that journal editorial will improve quality of the figures in the publication with pdfs provided.

Reviewer 2 Report

Comments and Suggestions for Authors

I read with great interest the paper entitled: “Evaluation of EBUS-TBNA Aspirates from Advanced NSCLC 2 for Comprehensive Sequencing Platforms Including Whole Ge-3 nome Sequencing.” This is a very well written manuscript. The introduction is concise and adequate. The authors showed that fresh EBUS-TBNA aspirates are appropriate for three sequencing platforms with comparable detection of driver mutations in specimens with ≥ 30% tumor amount. Comprehensive genomics bring benefits for patients where specimens present low DNA yield. The study seems very well designed and conducted, as well as the very accurate and smart choice of methods.

I would have some comments and suggestions:

Query 1. It is well known that conditions of prefixation, fixation, storage, and transportation of FFPE materials, together with post-fixation factors and the quality of laboratory materials used during extraction procedures significantly affect the quality of recoverable genetic material, especially when exiguous , therefore requiring adaptations to the protocol. These challenges highlight the need for alternative methods to obtain gDNA, including the use of slide scraping, aiming to improve the quantity and quality of the extracted material for application in sensitive genomic evaluation methodologies. Please specify how the FFPE specimens were processed by DNA extraction. Did you use scrapping, smear (in case of cytology aspiration) or sections of 5 micrometers? Please comment about the concordance or discrepancies between the two methods.

Query 2. How many specimens were represented by lymph node (with tumor metastases specimens) and how many were primary tumor mass?

Query 3. The cohort included cases of adenocarcinoma and squamous cell carcinoma. Usually, squamous cell carcinomas are central lesions in the bronchial tree, while adenocarcinomas are peripheral. Therefore, squamous cell carcinomas are more accessible to EBUS-TBNA. It would be interesting for the readers if the authors comment on the difference between the two histotypes in terms of mutation profile.

Query 4. The authors are kindly asked to provide a multi-perspective panel correlating H&E histological evaluation, Diff-Quik smear cytological evaluation involved estimation of the percentage malignant content of the smear, the malignant cell abundance (approximate number of malignant cells), the percentage area of the slide covered by smear and the DNA yield. In the same panel it is also important for the reader to include oncoplot of mutations detected in 40 patients across WGS, WES, TSO500 and standard of care (SOC), discrepancies of actionable mutations detected between sequencing platforms, and TMB.

Query 5. Please, comment about the concordance or discrepancies between histological and cytological specimens according sequencing platforms.

Query 6. The authors are kindly asked to provide comments about cost benefit.

Comments on the Quality of English Language

Overall,it is fine.

Author Response

Thank you for your time reviewing our manuscript.

Reviewer 2:

I read with great interest the paper entitled: “Evaluation of EBUS-TBNA Aspirates from Advanced NSCLC 2 for Comprehensive Sequencing Platforms Including Whole Genome Sequencing.” This is a very well written manuscript. The introduction is concise and adequate. The authors showed that fresh EBUS-TBNA aspirates are appropriate for three sequencing platforms with comparable detection of driver mutations in specimens with ≥ 30% tumor amount. Comprehensive genomics bring benefits for patients where specimens present low DNA yield. The study seems very well designed and conducted, as well as the very accurate and smart choice of methods.

I would have some comments and suggestions:

Query 1. It is well known that conditions of prefixation, fixation, storage, and transportation of FFPE materials, together with post-fixation factors and the quality of laboratory materials used during extraction procedures significantly affect the quality of recoverable genetic material, especially when exiguous, therefore requiring adaptations to the protocol. These challenges highlight the need for alternative methods to obtain gDNA, including the use of slide scraping, aiming to improve the quantity and quality of the extracted material for application in sensitive genomic evaluation methodologies. Please specify how the FFPE specimens were processed by DNA extraction. Did you use scrapping, smear (in case of cytology aspiration) or sections of 5 micrometers? Please comment about the concordance or discrepancies between the two methods.

Response: The technical aspects of specimen preparation for testing is important, as the reviewer highlights. The FFPE tissues mentioned in this study were only associated with standard of care testing by diagnostic pathology labs, which don’t use slide scraping. The clinical testing was recovered from medical records as samples were recruited from different hospitals and testing was performed in different pathology labs. The comparison across clincal labs was not in the scope of this study but is an important point. 

The EBUS-TBNA aspirates that were used for WGS/WES/TSO, were collected fresh frozen or in RNA later solution (see methods lines 129 and 144), not from scrapping FFPE sections or from smears such as Diff-Quik. We advocate the use of slide scrapings and have published on this previously; however, this was not the focus of the present study where we just focused on the outcomes from non-formalin-based samples (Frozen or in RNA Later).

The concordance between sequencing data from fresh EBUS samples and FFPE samples was in part discussed in the manuscript, but with a focus on comparing SOC testing (from FFPE) and research-based testing (from frozen/RNAlater), where we found good concordance for the detection of actionable mutations.

Tumour content of fresh samples was estimated using DNA and SNP arrays (Line 151) not by microscopic inspection of FFPE material.

Query 2. How many specimens were represented by lymph node (with tumor metastases specimens) and how many were primary tumor mass?

Response: We have included this information summarised in Table 1, and individual information was added to Table S2. In the cohort 8 cases were tumour mass, for an additional 38 cases the lymph node sampled was contiguous with the primary tumour mass.  

Query 3. The cohort included cases of adenocarcinoma and squamous cell carcinoma. Usually, squamous cell carcinomas are central lesions in the bronchial tree, while adenocarcinomas are peripheral. Therefore, squamous cell carcinomas are more accessible to EBUS-TBNA. It would be interesting for the readers if the authors comment on the difference between the two histotypes in terms of mutation profile.

Response: Figure S2 gives the readers some insight of the differences in mutations between the subtypes. We highlighted the point on Line 414. “Despite our small cohort, as expected these genes are frequently mutated in non-squamous NSCLC but not in the squamous subtype, highlighting the importance of future studies evaluating other lung cancer subtypes and that WGS of EBUS-TBNA could facilitate those discovery studies.”

Query 4. The authors are kindly asked to provide a multi-perspective panel correlating H&E histological evaluation, Diff-Quik smear cytological evaluation involved estimation of the percentage malignant content of the smear, the malignant cell abundance (approximate number of malignant cells), the percentage area of the slide covered by smear and the DNA yield. In the same panel it is also important for the reader to include oncoplot of mutations detected in 40 patients across WGS, WES, TSO500 and standard of care (SOC), discrepancies of actionable mutations detected between sequencing platforms, and TMB.

Response: In our earlier publication Fielding et al JTO Clin Res Rep. 2022 3(10):100403) we presented a detailed analysis of Diff-Quik smears, including a breakdown of cellular abundance and DNA content prior to sequencing. However, that was not the objective of this present paper and the smears were only used to guide specimen collection at bronchoscopy, where they were used to ensure an equitable sharing of the samples between standard of care and research samples. Diff-Quik smears were not sequenced in the present study. Histologic results for each patient were obtained from medical reports (H&E) and are presented as “Tumour Type” accompanying the molecular results for the 40 patients in the figures. Figure 2 and Table S3 present actionable mutations detected in 40 patients across WGS, WES, TSO500 and standard of care (SOC - FFPE) concordance and discrepancies are then discussed in the results section (See Lines 288 and 332). Comparison of TMB and other potential biomarkers of immunotherapy response across sequencing platforms (fresh tissue) are reported in Figure 4 and Figure S2).

Query 5. Please, comment about the concordance or discrepancies between histological and cytological specimens according sequencing platforms.

Response: Concordance and discrepancies of actionable mutations detected in clinical testing (FFPE) and fresh specimens is presented in section 3.3. Concordance and discrepancies across the 3 sequencing platforms for fresh specimens is discussed in section 3.3.1 (Lines 288 to 332). The tissue diagnosis (cell type) was only reported from the standard of care testing, presented in the figures as “Tumour Type).

Query 6. The authors are kindly asked to provide comments about cost benefit.

Response: Our sequenced cohort of 40 cases is underpowered for cost benefit estimations. In the discussion we had mentioned studies that have evaluated previously panel vs single gene testing. To our knowledge there is no cost benefit analysis of comprehensive sequencing using alternative sources of tissue for genomics in advanced lung cancers.  We have highlighted this point on page 16 Line 517.

“Previous studies evaluated the cost benefit associated with the implementation of comprehensive panels using FFPE specimens.  This does not address a current clinical limitation of insufficient material remaining for genomic testing after other diagnostic tests are performed. Unfortunately, our study is underpowered for cost benefit analysis. However, there is an urgent need to evaluate the cost benefit of introducing alternative sources of tumour tissue and different types of genomic testing in the clinical setting to improve the care of patients with advanced lung cancer.”

Reviewer 3 Report

Comments and Suggestions for Authors

The study by Fielding et al. investigates an important issue in NSCLC biomarker testing: the small-biopsy problem. Or, in other words, the ever-shrinking biopsy (often, as in this case, limited to cytologic material) and the ever-growing number of actionable genetic alterations to test, especially when testing can be performed with multiple techniques.

The authors perform the largest study (to date and to the best of my knowledge) on fresh EBUS-TBNA aspirates, assessing their suitability for massive genomic analysis with multiple platforms, in addition to standard-of-care testing.

The Abstract and simple summary are short and informative. The structure of the text is clear. The table is necessary and informative.

I have no major suggestions.

Minor suggestions:

The English language is flawless, but there are rare typographical errors (e.g. L59: "play" should be "plays"; Table 1, the "Never Smoked" percentage has two decimal points).

The figures are hardly readable in the pdf version of the manuscript. Online they are of course fine.

I would have liked a slightly longer discussion (in the intro or discussion, e.g. line 482) on previous experience on biomarker testing in NSCLC on cytologic material (especially fresh, more than fixed (e.g. scraped from slides)), mentioning for example 10.1016/j.prp.2021.153547 and 10.1016/j.jtocrr.2020.100077

Author Response

Thank you for your time reviewing our Manuscript.

We have added text to improve the introduction: See Line 66: “There are clinical challenges with the increasing requirement for more molecular diagnostic tests and the small biopsy specimens available for analysis. There is a need to evaluate methods to could fast track more comprehensive genomic testing to improve care of patients with advanced lung cancer.”

The study by Fielding et al. investigates an important issue in NSCLC biomarker testing: the small-biopsy problem. Or, in other words, the ever-shrinking biopsy (often, as in this case, limited to cytologic material) and the ever-growing number of actionable genetic alterations to test, especially when testing can be performed with multiple techniques.

The authors perform the largest study (to date and to the best of my knowledge) on fresh EBUS-TBNA aspirates, assessing their suitability for massive genomic analysis with multiple platforms, in addition to standard-of-care testing.

The Abstract and simple summary are short and informative. The structure of the text is clear. The table is necessary and informative.

I have no major suggestions.

Minor suggestions:

The English language is flawless, but there are rare typographical errors (e.g. L59: "play" should be "plays"; Table 1, the "Never Smoked" percentage has two decimal points).

Response: We have corrected the typographical errors above.

The figures are hardly readable in the pdf version of the manuscript. Online they are of course fine.

I would have liked a slightly longer discussion (in the intro or discussion, e.g. line 482) on previous experience on biomarker testing in NSCLC on cytologic material (especially fresh, more than fixed (e.g. scraped from slides)), mentioning for example 10.1016/j.prp.2021.153547 and 10.1016/j.jtocrr.2020.100077

Response: We have incorporated the papers suggested and have extended the discussion to address this and other reviewers’ points. See line 497.